# The Dilemma of Balance between Benefits and Losses for Chondromyxoid Fibroma-like Osteosarcoma in Thoracic Spine with Azygos Tumor Thrombosis: A Case Report

**DOI:** 10.3390/medicina59040792

**Published:** 2023-04-19

**Authors:** Chuanchao Du, Xiaoguang Liu, Zhongjun Liu, Feng Wei

**Affiliations:** 1Department of Orthopedics, Peking University Third Hospital, Beijing 100191, China; 2Department of Orthopedics, Emergency General Hospital, Beijing 100028, China

**Keywords:** primary spine tumor 1, chondromyxoid fibroma-like osteosarcoma 2, azygos vein with tumor thrombus 3

## Abstract

*Background*: Chondromyxoid fibroma-like osteosarcoma (CMF-OS) is an extremely rare subtype of osteosarcoma, its clinical data are scarce, and our understanding of it is far from sufficient. As it has few typical imaging manifestations, it is not uncommonly misdiagnosed clinically. Azygos vein thrombosis is also a rare entity, and there is a big controversy over treatments for it. *Case presentation*: Herein, we report a case of CMF-OS that occurred in the spine, coincidently, azygos vein thrombosis was found. A young male patient came to our clinic because of continuous back pain, and a neoplastic lesion was suspected in the thoracolumbar vertebrae. The pathological results of the biopsy showed a low grade of osteosarcoma, and chondromyxoid fibroma-like osteosarcoma was the primary diagnosis. Since the tumor cannot be en-bloc resected, he received palliative decompression surgery, followed by radio and chemotherapy. Azygos vein tumor thrombosis was not treated and, unfortunately, he died of heart failure caused by the thrombus migrating from the azygos vein to the right atrium. Before the palliative decompression surgery, both the patient and the clinical team were trapped in the dilemma of how big a surgery should be carried out to maximize the benefits of this patient. *Results and complications*: CMF-OS is indeed more aggressive than its pathological sections suggest. Guidelines for osteosarcoma should be followed. Furthermore, it is important to recognize the danger of tumor thrombosis in the azygos vein. Preventive measures have to be performed in a timely manner to avoid catastrophic results.

## 1. Introduction

Chondromyxoid fibroma-like osteosarcoma (CMF-OS) is an extremely rare type of low-grade osteosarcoma [1]. It has the imaging and pathological manifestations of osteosarcoma-like focal osteogenesis. A lot of myxoid stroma is usually found among loose aggregated cells, which are separated by fibrovascular septa. The tumor cells usually have a low grade of atypia and mitotic phase [2]. However, CMF-OS is more biological aggressive than it appears, and the prognosis is commonly poor [2]. Because of its atypicality, misdiagnosis is more likely to occur. Here, we report a case of CMF-OS involving extensive thoracolumbar (T10-L1) vertebrae and paravertebral tissues. Worse still, azygos vein tumor thrombosis was found, and its diagnosis and treatments were little known [3]. The young patient and the treatment team were entangled in the benefits and losses that surgery can bring. This is a dilemma that exists widely between doctors and patients.

## 2. Case Report

A 17-year-old male patient came to our clinic with back pain for 7 months with movement restriction in the morning, while symptoms were usually partially relieved 1 h after exercise. In the beginning, he was diagnosed as having “kidney stones” in a local hospital, and was given lithotripsy. However, his lower limbs became progressively weaker during the following 3 months, and he had to obtain assistance when climbing up stairs. Physical examinations found that more than half of the key muscles’ strength of the lower limbs were 4/5 (American Spinal Cord Injury Association, ASIA D) [4].

Lumbar X-ray films undertaken in a local hospital showed signal changes in the T11 vertebra, which was whiter on the X-ray film (Figure 1A,B); furthermore, CT scan images demonstrated osteogenic lesions in the vertebrae and the appendices of T11 and T12 (Figure 1C,D; Figure 3A). MRI scan images further showed high signal changes in the vertebrae both in T1-weighted and T2-fatty suppressed images, pars and paravertebral tissue from T11 to L1, and, furthermore, the spinal canal was partially occupied and the spinal cord was compressed (Figure 2A,B; Figure 3B). Then, a needle biopsy was undertaken at the L1 vertebra under CT navigation, and CMF-OS was the primary diagnosis. A high metabolic level was found in the above sites (including the azygos vein at T11 level) in systemic metabolic nuclide images. No previous medical history was declared.

### 2.1. Pathology Report

The T11 paravertebral soft tissue was basically highly calcified myxoid stroma tissues with a disordered structure; tumor cells were sparsely distributed and separated by cells rich in sheath. The vertebral bone tissue was partially replaced by multi-nodular lesions. The myxoid stroma can be seen in the lesions with focal calcification and osteogenesis, and the peri-nodular interstitial cells were rich and nuclei had heteromorphism. From the results of cell morphology and immunohistochemistry, chordoma (usually having a bubbly/vacuolated appearance) was excluded, and chondromyxoid fibroma-like osteosarcoma was considered, because of the presence of osteoid production directly by the tumor cells under a chondromyxoid fibroma (CMF)-like background (Figure 4A,B).

For the immunohistochemistry results, in detail, S-100 (−), SOX9 (−), P53 (−), Osteopontin (OPN, +), Osteonectin (+), CD34 (−), Ki67 (6%), brachyury (−), vimentin (+) (Figure 4C,D). OPN can be secreted by osteoblasts, osteocytes, and osteoclasts, which plays an important role in the mineralization and resorption of bone matrix. OPN is abundant in endochondral bone and endomembranous bone, and can be observed in the cytoplasm of osteoblasts and osteocytes in woven bone [5]. OPN is overexpressed in many malignant tumors, including breast cancer, lung cancer, gastric cancer, ovarian cancer, and melanoma. Recent studies suggest that OPN may play an important role in the diagnosis and treatment of osteosarcoma [5]. Osteonectin (ON), a 32,000-kd glycoprotein involved in the early steps of Osteonectin (ON) mineralization in bone tissue, is a recognized differentiation marker of normal osteoblasts. It is helpful for the histological diagnosis of bone tumors, especially for the differentiation of small cell osteosarcoma from other small round cell tumors [6].

From the pathology report, chondromyxoid fibroma-like osteosarcoma was considered a kind of low-grade osteosarcoma. The tissue slides were used in consulting with several well-known pathologists from several hospitals, and the impression was consistent with the primary diagnosis.

### 2.2. Treatment

Chemotherapy was firstly considered for osteosarcoma to eradicate the malignant cells in circulation, reduce the tumor size, and create opportunities for tumor resection [7]. The patient received two courses (2 months) of chemotherapy with methotrexate (16 g) and vincristine (2 mg).

During the course of chemotherapy, the neurological status of the lower limbs further deteriorated, and the muscle strength dropped to 0 grade, while the function of defecation and urination was normal. After multidisciplinary consultation, it was recommended that surgery must be performed emergently to save nerve function, and then chemotherapy or other treatments should be continued.

Preoperative physical examination found that pinprick and temperature sensation in the skin below the groin had disappeared, while proprioception sensation existed. Muscle strength of the lower limbs below the iliopsoas muscle was 0/Ⅴ grade. Hyperreflexia was found in both patellar tendons and the Achilles tendon; the Babinski sign was negative, while ankle clonus was positive on both sides. Based on the physical examinations, the spinal cord injury was incomplete (ASIA B).

Before operation, the selected vertebral arteries were thrombosed by vascular intervention surgeons. A traditional posterior approach was used, and a laminectomy was chosen from T7 to L4. Then, pedicle screw fixation was implanted for the vertebrae of T7, 8, 9 and L2, 3, 4, which are the three vertebra segments above and below across the lesion site. Intralesional resection was executed for T10, 11, 12, and L1 (lamina, upper, and lower articular processes and pedicles), and the tumor in the intraspinal canal was also resected; however, there was still much tumor tissue remaining in the paravertebral and vertebra region. The operation took 5 h, and the bleeding volume was 1500 mL. Post-operative X-ray films are seen in Figure 5.

Nerve function recovered partially, e.g., iliopsoas, quadriceps, and other key muscle strengths recovered to level Ⅰ/Ⅴ. The general condition was good and the nutritional status was basically normal (albumin 39 g/L, hemoglobin 107 g/L). Post-operative pathology confirmed the diagnosis of CMF-OS, but tumor cells did not proliferate fast, therefore, the patient was not suitable for high-throughput genetic testing. Furthermore, no obvious tumor necrosis was found after chemotherapy, which verified that the tumor was not sensitive to chemotherapy of the MTX and vincristine.

The final clinical course was chemotherapy, which may not be a satisfactory method for CMF-OS; radiotherapy should be preferentially considered. More importantly, azygos vein thrombosis should be given close attention in the case of fatal heart-pulmonary embolism. For this tough situation, multi-disciplinary treatment (MDT) should be considered, e.g., minimal invasive vessel intervention therapy (vessel stent) may be used. Furthermore, more aggressive interventions such as azygos vein excision may be considered to prevent fatal complications.

## 3. Discussion

### 3.1. CMF-OS, Low-Grade, but Highly Aggressive

CMF-OS is an extremely rare type of malignant tumor, and shares the same biological behavior as osteosarcoma, so it is defined as a subtype of osteosarcoma and follows the same principles of treatment [2]. After the medical history collection and physical examination, the regional lesion was initially evaluated by CT and MRI, and distal metastasis was excluded by Pet/CT. Furthermore, bone metabolism should also be tested to evaluate bone metabolism [7].

For low-grade osteosarcoma, it is well established that extensive resection should firstly be considered, followed by sensitive chemotherapy or radiotherapy if there is residual tumor tissue. Although first line chemotherapies for osteosarcoma are cisplatin and doxorubicin with or without a high dose of methotrexate, obviously, the regular chemotherapy plan is not fit for a low proliferative osteosarcoma such as CMF-OS, and the poor results of chemotherapy for this patient confirmed this viewpoint. Methotrexate plays an anti-tumor role mainly by inhibiting the DNA synthesis of tumor cells [8]. The vincristine targets microtubules, which mainly inhibit the polymerization of tubulin, affect the formation of spindle microtubules, and finally stop mitosis in the metaphase [9]. The reasons for choosing MTX and vincristine, which are experimental treatments, include their relatively low toxicity. However, after two courses of chemotherapy, the neurological status deteriorated very quickly, and palliative decompression had to be carried out urgently to preserve neural function.

According to the NCCN guideline, radiotherapy should be considered for patients with positive margins of resection, partial resection, or unresectable cases [10]. This patient received a large fractionation radiotherapy of total 60 Gy. Until he passed away (6 months after the operation), no local or distant recurrence was found, which may be partially due to the effect of radiotherapy.

### 3.2. Proactive Treatments Are Suggested for Azygos Vein with Tumor Thrombosis

The azygos vein is usually located at the right anterior side of the spine (from T1 to L2), it connects with the right subcostal vein, inferior vena cava, ascending lumbar vein, accessory hemiazygos vein, and intercostal veins. It mainly hosts the venous return of the chest wall and posterior mediastinal organs, and is one of the important connecting channels between the superior and inferior vena cava. When the superior vena cava or inferior vena cava blood flow is blocked, these anastomosis channels become an important pathway of collateral circulation. Infection of the thoracoabdominal cavity can spread to the skull and vice versa due to the communication of the azygos vein with the intravertebral and external venous plexus which in turn communicate with the intracranial venous sinuses. During en-bloc spondylectomy of thoracic vertebrae, attention should be paid to this venous system, because accidental injury of the venous system may lead to massive bleeding [11]. For this case, medical images showed venous thrombosis in the azygos vein; however, concerning the big risk of surgical intervention with this tough situation, conservative treatments were executed. Unfortunately, without intervention, the patient died from heart failure caused by thrombus in the atrium which probably came from the azygos. Therefore, resection of the azygos with thrombosis should be actively considered in case of migration to the atrium. As far as we know, this is the first report on heart failure caused by azygos tumor thrombus.

### 3.3. The Dilemma of Benefits and Loses

At the beginning, tumor tissue should be resected as much as possible, including vertebrae, paravertebral tumor tissues, and the azygos vein with tumor thrombosis. Theoretically, two stages were needed to finish the operation. Firstly, the azygos vein with thrombosis should be resected, and tumor tissue in the anterior and lateral side should be freed through an anterior approach. Secondly, 3D-printed prosthesis vertebrae or a titanium mesh can be used to achieve spine reconstruction through a posterior approach. However, tumor tissues were so extensive that it was hard to complete en-bloc resection. The possibility of complete resection was pretty low, and the complication rate was very high (hemorrhagic shock, deep infection, deterioration of nerve function, implant failure, et al.), placing his life in danger. Therefore, en-bloc resection was very challenging, and intralesional resection was chosen to preserve neurological function and decrease perioperative complications.

Furthermore, tumor thrombosis was found in the azygos vein, which is also a rare phenomenon and difficult to treat. The question was whether we should perform an excision of the azygos vein with thrombosis at great risk of major complications. Even though it is a major operation [3], the resection of the azygos vein with thrombosis may save his/her life because thrombus formed in the azygos vein may migrate upward, and finally result in superior vena cava and heart thrombosis [12]. However, the tumor could only be partially resected and life expectancy was poor—both the patient and surgeons did not want to take the risk of resection of an azygos vein with thrombosis. In addition, a venous stent may be applied to recanalize the azygos vein by vascular and interventional radiologists [13].

Furthermore, a poor prognosis was deduced considering the unresectable tumor and poor response to neoadjuvant chemotherapy. Based on the above issues, palliative resection of the tumors by a posterior approach was chosen with the main purpose of saving nerve function. Then, the chemotherapy plan was readjusted according to the postoperative pathology results; similarly, whether to remove the anterior vertebral body depended on the postoperative pathology report and the advice from the multidisciplinary consultation. 

The balance of benefit and loss was hard to achieve when we just considered the state of illness, let alone the very poor financial compensation he could receive. However, it is common to see families suffer extreme poverty because of serious diseases in many countries. Furthermore, life expectancy is not always easy to judge according to tumor classification and stage. All in all, it is difficult to balance possible complications, benefits, and the financial burden of challenging operations.

Finally, this patient received palliative decompressive surgery. Unfortunately, he died of heart failure caused by thrombus from the azygos vein, which was not anticipated. In other words, resection of the azygos vein with thrombosis in time may have saved his life. This was a bitter lesson, worthy of our attention.

## 4. Conclusions

In this report, a case with spine CMF-OS was reported, and it was much more malignant than its pathology suggested. The patient did not respond well to chemotherapy, and the prognosis was very poor. Furthermore, it should be emphasized that azygos vein thrombosis was present, and the patient died of heart failure caused by thrombus in the right atrium. Therefore, we recommend active intervention to deal with the azygos vein thrombosis and prevent thrombus from upward migration.

## Figures and Tables

**Figure 1 medicina-59-00792-f001:**
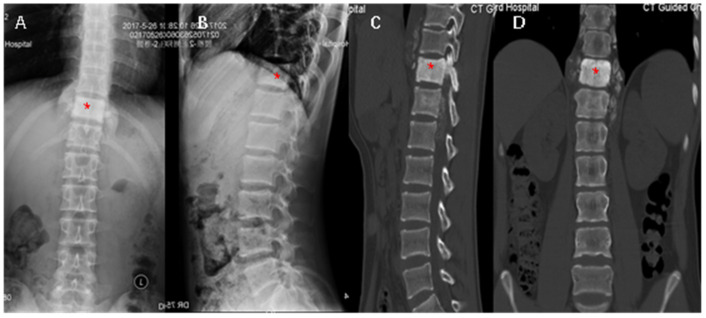
X-ray films (**A,B**) and CT images (**C**,**D**) showed the high-density osteogenic changes (* in red) in the T11 vertebrae body (**A**–**D**) and appendages (**C**,**D**), and those involved with the upper and lower stages of the spinal canal (**C**,**D**).

**Figure 2 medicina-59-00792-f002:**
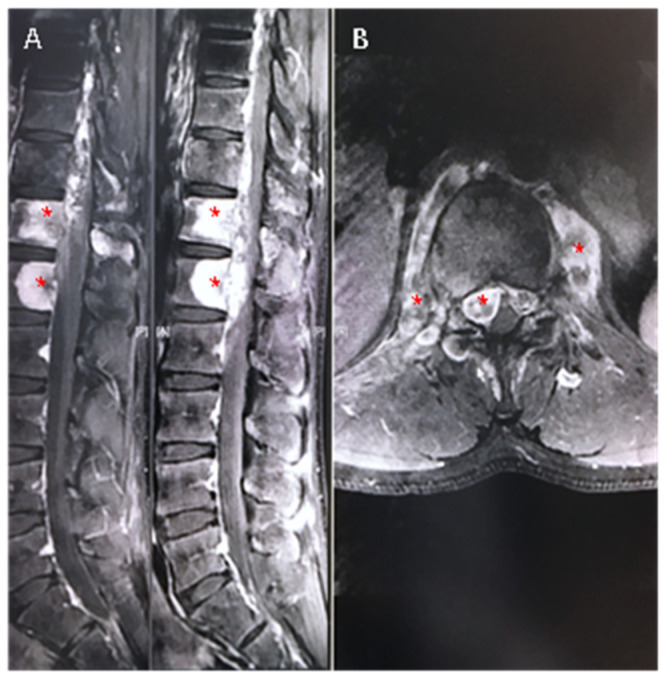
Abnormal signals were found in T11, T12 and the corresponding spinal canal in contrast-enhanced magnetic resonance images both in sagittal (**A**) and axial images (**B**) (T1-weighted imaging). Tissues suspected of being tumors can be significantly enhanced in MRI images (* in red). Part of the spinal canal was occupied by the lesion and the dura sac was obviously compressed.

**Figure 3 medicina-59-00792-f003:**
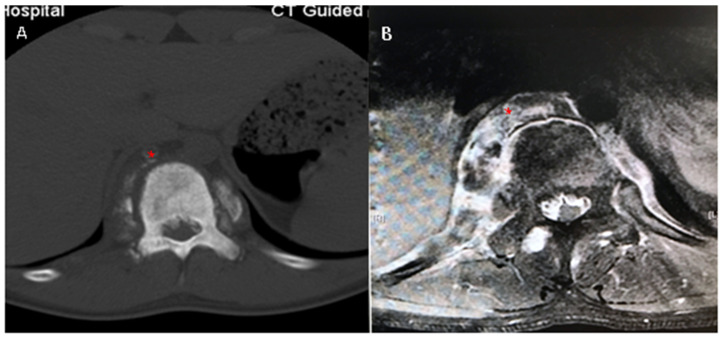
The azygos vein (* in red) was occupied by tumor-like tissues (**A**), and the tissues were significantly enhanced in MRI T1-weighted imaging (**B**).

**Figure 4 medicina-59-00792-f004:**
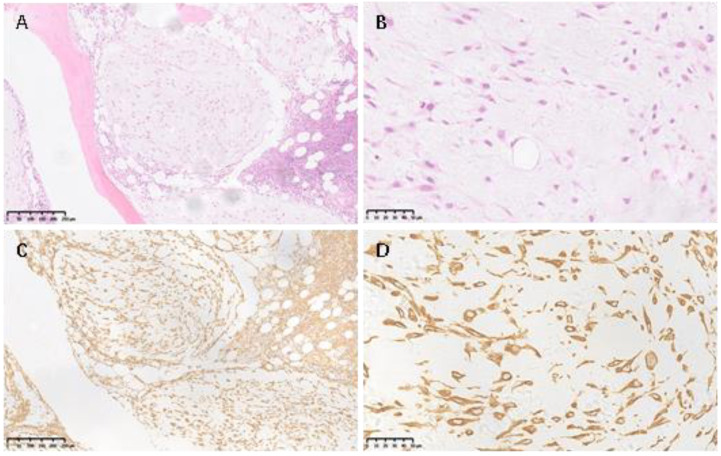
Hematoxylin–eosin staining of T11 paravertebral tissue showed nodular calcification and osteogenesis separated by cells rich in septa (**A**, 100×). Under high-power microscopy (**B**, 400×), tumor cells were loosely aggregated in myxoid stroma and osteoid was produced, and nuclei heteromorphism can be found. Vimentin staining of the same section was positive (**C**, 100×; **D**, 400×).

**Figure 5 medicina-59-00792-f005:**
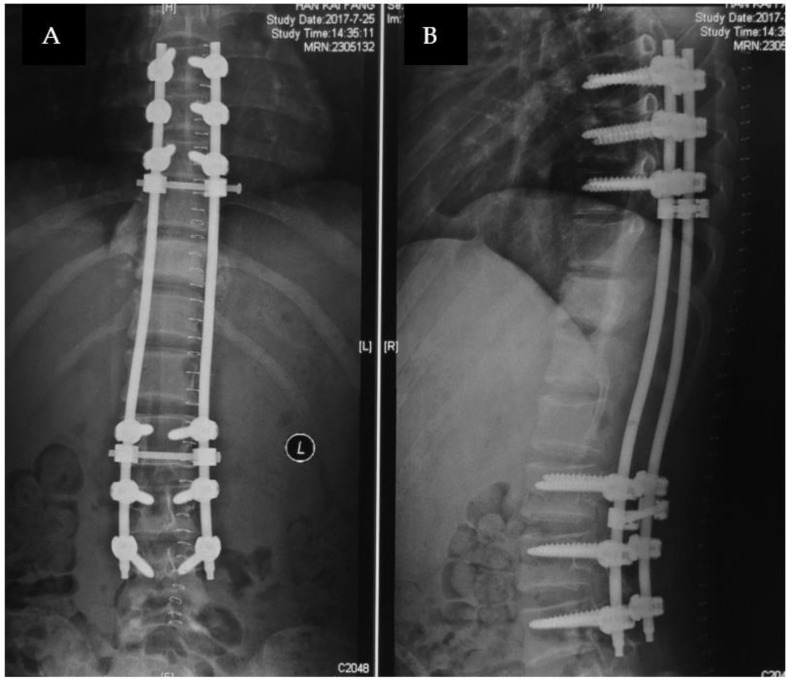
Anterior−posterior (**A**) and lateral X-ray (**B**) after palliative decompression surgery of the patient with CMF-OS.

## Data Availability

Not applicable.

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
