# Peer review of "The Dilemma of Balance between Benefits and Losses for Chondromyxoid Fibroma-like Osteosarcoma in Thoracic Spine with Azygos Tumor Thrombosis: A Case Report"

_medicina, 2023, doi:10.3390/medicina59040792_

Round 1

Reviewer 1 Report

Dear Editor,

Thank you for giving me an opportunity for reviewing the manuscript.

This manuscript is about challenging treatment for Chondromyxoid fibroma-like osteosarcoma (CMF-OS) invading the adjacent tissues including azygos vein.

I understand this is a difficult case, but treatment option seems inappropriate for impending paralysis due to unresectable spine malignant bone tumor.

My first impression is that this manuscript is not acceptable for publication because of unreasonable treatment approach.

Major issues

1.      The authors recognized limb muscle weakness (MMT 4/5). Moreover, images clearly showed compression of spinal canal/cord. Impending paralysis was obvious at this moment. In parallel, CT-guided biopsy indicated low grade CMF-OS which was unresectable in spine. Therefore, urgent radiotherapy with or without spinal decompression surgery was the first choice, whereas chemotherapy was not. In my opinion, first treatment option seems inappropriate.

2.      If chemotherapy was used from the authors’ thought, I would like them to present previous papers about good effectiveness of chemotherapy for CMF-OS because basically chemotherapy should not be used for low grade OS.

Author Response

Dear reviewer,

Thank you for your excellent comments. This manuscript is about a tough rare case about CMF-OS. Because this is an extremely rare disease, many spine surgeons are not aware of the characteristics. So from this viewpoint. The meaning of this manuscript is a good reference for readers. Secondly. it's true most of low grade osteosarcoma are not sensentive to chemotherapy. However, it is not certain that CMF-OS is low grade of osteosarcoma. Because of poor Knowing of this rare case, trial Chemotherapy was applied, and the response was poor. Even though the patient deteriorated after two periods of chemotherapy,which indicated poor responses of chemotherapy. Who says a failure case can't be published?

thanks again,

yours Dr.Du

Reviewer 2 Report

General impression

In this case presentation, the authors reported the patient with chondromyxoid fibroma-like osteosarcoma (CMF-OS) accompanied by thrombosis in azygos vein.  As they mentioned, CMF-OS is an extremely rare type of malignant tumor lacking sufficient clinical experiences, so that it must be needed s special management in each case.  I evaluate this paper includes valuable clinical information to announce the spine surgeons.

For these reasons, I think this manuscript is appropriate for publication.

However, I have a couple of requests to be revised as stated below.  After they have been resolved, I will judge this manuscript can be accepted and published by medicina journal.

1. Introduction, page 1 line 30

  I recommend the full term of CMF-OS; “chondromyxoid fibroma-like osteosarcoma” is indicated here because it is firstly seen in the text.

“CMF-OS”will be replaced by “Chondromyxoid fibroma-like osteosarcoma (CMF-OS)”.

2. page 3 line 75 and 79

The words of “chondromyxoid fibroma-like osteosarcoma” in these places can be replaced by “CMF-OS”.

3. Discussion, page 4 line 118

  “bone metabolism, e.g.,” between “Besides,” and “ALP,” can be deleted because it was repeated at the end of sentence.

4. The final clinical course should be added in the end of “Case report section”.

5. postoperative x-rays

  I recommend postoperative x-rays (a-p and lateral views) will be shown if available.

Author Response

Dear Reviewer,

Thank you so much for your excellent comments. I have revised the manuscript following your suprevision.

By the way, the description of the pathology report was totally consistent with the ideas of pathologists.The myxoid stroma or chondromyxoid fibroma are just description of pathology view.

Besides, the postoperative films will be uploaded after informed consent from the patient.

yours sincerely

Dr.Du

Round 2

Reviewer 1 Report

More references about CMF-OS are needed If the authors insists that This manuscript is about a tough rare case about CMF-OS. Because this is an extremely rare disease, many spine surgeons are not aware of the characteristics. Secondly. it's true most of low grade osteosarcoma are not sensentive to chemotherapy. However, it is not certain that CMF-OS is low grade of osteosarcoma.

I am aware of the rarety of CMF-OS. In discussion part, only citation 2 is what the authors explained about CMF-OS. Apparently, this is too small to judge the importance of this case report. Much more information is critically required. 

Thanks,  

Author Response

Dear Reviewer,

Thank you so much for your excellent comments. I have revised the manuscript following your suprevision.

 I totally agreeed with you that it's true most of low grade osteosarcoma are not sensentive to chemotherapy,just like what you said it is not certain that CMF-OS is low grade of osteosarcoma.

It is a tough case to deal with, on the one hand is extremely rare, on the other hand, azygos venous thrombosis bringed a bigger challenge to us, both of them indicated the siginificantness of this case.

Further more, this manuscript provided many images, and treatments details, which all in all this manuscript will bring a lot learning to readers.

yours sincerely

Dr.Du